# Developing a strategy to scale up place-based arts initiatives that support mental health and wellbeing: A realist evaluation of 'Arts for the Blues'

**Vicky Karkou**[1]*, **Joanna Omylinska-Thurston**[2,3], **Scott Thurston**[4], **Rebecca Clark**[5], **Emma Perris**[6], **Axel Kaehne**[7], **Mark Pearson**[8]

**1** Research Centre for Arts and Wellbeing, Health Research Institute, Faculty of Health, Social Care and Medicine, Edge Hill University, Ormskirk, Lancashire, United Kingdom, **2** School of Health and Society, University of Salford, Friedrick Campus, Salford, United Kingdom, **3** Counselling Psychologist at Greater Manchester NHS Foundation Trust, Chorlton-Cum-Hardy, Manchester, United Kingdom, **4** School of Arts and Media and Creative Technology, University of Salford, Salford, Greater Manchester, United Kingdom, **5** School of Health and Society, University of Salford, Salford, Greater Manchester, United Kingdom, **6** Research Centre for Arts and Wellbeing, Edge University, Ormskirk, Lancashire, United Kingdom, **7** Evaluation and Policy Analysis Unit, Edge Hill University, Ormskirk, Lancashire, United Kingdom, **8** Institute for Clinical & Applied Health Research, Hull York Medical School, University of Hull, Hull, United Kingdom

* karkouv@edgehill.ac.uk

**Data Availability Statement:** The full dataset is available on figshare under the following DOI: https://doi.org/10.25416/edgehill.24456367.v1.

## Abstract

Place-based arts initiatives are regarded as rooted in local need and as having capacity to engage local assets. However, many place-based arts initiatives remain poorly funded and short-lived, receiving little attention on how to *scale up* and *sustain* their activities. In this study we make a unique contribution to knowledge about scaling up place-based arts initiatives that support mental health and wellbeing through focusing on the example of 'Arts for the Blues', an arts-based psychological group intervention designed to reduce depression and improve wellbeing amongst primary care mental health service users in deprived communities. Methodologically, we used realist evaluation to refine the study's theoretical assumptions about scaling up, drawing on the lived and professional experiences of 225 diverse stakeholders' and frontline staff through a series of focus groups and evaluation questions at two stakeholders' events and four training days. Based on our findings, we recommend that to scale up place-based arts initiatives which support mental health and wellbeing: (i) the initiative needs to be adaptable, clear, collaborative, evidence-based, personalised and transformative; (ii) the organisation has to have a relevant need, have an understanding of the arts, has to have resources, inspiration and commitment from staff members, relevant skillsets and help from outside the organisation; (iii) at a policy level it is important to pay attention to attitude shifts towards the arts, meet rules, guidelines and standards expected from services, highlight gaps in provision, seek out early intervention and treatment options, and consider service delivery changes. The presence of champions at a local level and buy-in from managers, local leaders and policy makers are also needed alongside actively seeking to implement arts initiatives in different settings across

**Funding:** 2022-01-31 to 2023-01-30 | Grant Arts and Humanities Research Council (Swindon, GB) GRANT_NUMBER: AH/W007983/1 The funders had no role in study design, data collection and analysis, decision to publish, or preparation of the manuscript.

**Competing interests:** The authors have declared that no competing interests exist.

geographical spread. Our theoretically-based and experientially-refined study provides the first ever scaling up framework developed for place-based arts initiatives that support the mental health and wellbeing, offering opportunities for spread and adoption of such projects in different organisational contexts, locally, nationally and internationally.

## Introduction

Depression is one of the most common mental health concerns and a world-wide problem affecting an estimated 5% of adults [1]. In the UK, the number of adults experiencing depression has doubled since before the pandemic: in early 2021, around 1 in 5 (21%) people experienced some form of depression ranging from mild to severe [2], with young people and women being affected the most. Public Health England estimates for 2019/20 indicate that this is part of a longstanding and wider problem with almost 17 percent of people in the UK being affected by common mental health problems, including not only depression but also generalised anxiety disorder, panic disorder, phobias, obsessive-compulsive disorder and post-traumatic stress disorder.

Poor mental health is often associated with equally poor overall health and exacerbated by health inequalities and socio-economic deprivation. The North West of England for example, has some of the highest levels of mental health problems in the country [3] and, at the same time, the highest levels of unemployment and economic deprivation [4]. The problem has worsened since the Covid-19 pandemic with the latest Marmot Review [5] suggesting that the amount of time people spend in poor health has increased since 2010. The same report also points out that the increase in poor health also means an increase in spending from the 'public purse' (p.3).

Psychological support for common mental health problems in the UK is mainly offered by National Health Service (NHS) Talking Therapies (formally known as Improving Access to Psychological Therapies [IAPT]) [6]. NHS Digital [3] reports that only around half of those attending these services (46.4%) complete the therapies on offer. It is clear that there is an urgent need to rethink current health and mental health provision in England, making it more attractive and engaging to a wider range of people.

High dropout rates from current statutory mental health provision may be due to heavy reliance on talking therapies. The most common amongst them is Cognitive Behavioural Therapy (CBT) [3] an intervention that focuses on cognitive processing, assumes good use of English and a 'good enough' British-based educational background. These assumptions can be problematic in communities that struggle with health inequalities and economic deprivation [7, 8]. The extensive use of CBT in primary mental health services has also raised concerns around its value for Black, Asian and Minority Ethnic (BAME) communities [9], refugee and migrant communities [10], people with disabilities [11] and so on. The need for additional provision is clear.

The charity MIND identified arts-based approaches as one of their clients' top preferences for psychological support [12], whilst clients in deprived areas also reported their preference for more creative interventions [13]. More recently, a survey completed by Millard et al. [14] that included 1541 participants (685 mental health patients and 856 members of the general population) showed that 60% were very interested in group arts therapies. However, non-verbal, creative, arts-based interventions such as therapies are rarely used in the North West of England in statutory services. At the same time, available community arts activities in the

region require support, robust training and evaluation in order to achieve and capture intended mental health outcomes.

The Marmot Review [5] argues that health inequalities are "unnecessary and can be reduced with the right policies" (p 3). This calls for a response, which, given the complexity of health inequalities in the region, requires a careful consideration of useful, place-based interventions from the local level right up to a systems level. In this paper we report on how we scaled up an arts-based group intervention called 'Arts for the Blues' in the North West of England. The intervention was originally designed to address the needs of users of primary mental health services and specifically NHS Talking Therapies services. We aimed to offer an additional option to service users that could tackle depression in deprived communities in inner city Manchester [15]. Since then, we have piloted the work with adults in local mental health charities and with children in school settings in the North West. This evidence-based intervention draws on a systematic review of helpful factors of National Institute for Health and Care Excellence (NICE)-recommended psychotherapies and arts therapies [16], creative explorations in the studio [17] and several workshops with staff and service users [18].

The Arts for the Blues intervention is based on a pluralistic approach [19] and offers a structure based on eight key ingredients [20] which are delivered through creative means such as visual arts, dance, drama, music and creative writing. The eight ingredients are: (i) active engagement, (ii) learning skills, (iii) developing relationships, (iv) expressing emotions, (v) processing at a deeper level, (vi) gaining understanding, (vii) experimenting with different ways of being and (viii) integrating useful material. So far it has been delivered to groups of 4–9 participants over 12 sessions, drawing on Yalom's group theory [21]. In all cases the use of creative methods is regarded as an important factor in achieving health outcomes (www.artsfortheblues.com).

For both adults and children, the evidence so far offers promising results. For example, during 12 sessions of delivery, adults with depression who received support through the mental health charity MIND, not only reduced symptoms of depression (PHQ9) but also reduced anxiety (GAD7) and increased scores of wellbeing (WHO-5) (Omylinska-Thurston et al., in preparation). In other community settings we found that psychological flow improved and was linked with improvements in mood (PANAS) and goal attainment [22]. In our work with children [23], we found that there was an overall reduction of emotional and behavioural difficulties (SDQ). More specifically, in a pilot randomised controlled study with 62 children aged 9–11 attending primary schools in the North West [23], life functioning (CORS) was improved after eight sessions with significant differences from the waiting list control group. This change was sustained over a year, whilst the quality of children's sleep improved as indicated by scores gained by activity watches. Positive results were also found in our study with 37 caregivers of children with autism [24] with caregivers showing minimal clinically important differences in their scores on wellbeing (Adult Wellbeing Scale–AWS) and stress (Parenting Stress Index-Short Form–PSI-SF) after five sessions.

The above findings resonate with findings from Cochrane Reviews in arts-based interventions [25, 26] and other systematic reviews on depression [27–31] indicating that, despite the small sample size and methodological limitations of the included studies, when arts therapies are added to standard care, they are more effective than standard care alone in reducing levels of depression.

The value of the arts in tackling depression, improving wellbeing and supporting aspects of social cohesion is reported in the All-Party Parliamentary Group on Arts, Health and Wellbeing [32], as well as more recently in the World Health Organisation (WHO) scoping review [33] and evidence summary report to the Department of Digital Culture, Media and Sports [34]. Still, the integration of creative practice in health systems remains limited, more so in deprived areas such as the North West, as is the integration of learning gained from place-based initiatives such as the Arts for the Blues model.

## Scaling up complex interventions

Scaling up complex interventions in health has been extensively discussed by researchers and policy makers. A recent study by Willis et al. [35] used a realist synthesis approach to investigate which factors contributed to the success of scaling up initiatives in the health sector. The authors found that four broad domains influenced the probability of health interventions being implemented on a wider scale: awareness, commitment, confidence and trust. In each domain, specific actions were responsible for triggering positive circumstances that promoted the adoption and spread of health interventions in new settings, such as strong partnerships, the willingness to engage in evaluating their interventions, significant political support and the existence of adaptive funding models [35].

The Academic Health and Science Network (AHSN) also contributed significantly to our thinking around spread and adoption [36]. They argue that for an innovation to be spread and adopted, it needs to be simple, flexible and adaptable. At the same time, the report acknowledges that healthcare organisations are complex, and as such they may lead to complex processes of adoption.

The most common strategies to spread and adopt innovations within healthcare include (i) models that support quality improvement, (ii) cohesive frameworks from the beginning to the end, (iii) project management approaches that are informed by implementation science and (iv) coaching approaches that focus on behavioural change.

A recent rapid review of the practice of spreading health care innovations also indicated that practitioners used a variety of different approaches to support the adoption and scaling up of new initiatives or programmes [37] and reduced these approaches down to three perspectives, those stemming from implementation science, complexity science and social science:

- implementation science championed structured changes as part of quality improvement cycles in health services;

- complexity science perceived scaling up as an adaptive change occurring over time which required dedicated supportive actions;

- Social science saw scaling up health interventions as part of the social dynamics between stakeholders, often drawing on social action theory.

The authors conclude that successful scaling up approaches were utilising at least one of the models above, whilst those failing to theorise on behaviour, strategies and actions through any theoretical lens were unlikely to succeed in scaling up their intervention in health services [37].

Our own study drew on wider implementation theory [38–40] falling at the intersection of implementation and complexity science. As a result, we were interested in both actively bringing about quality improvement changes as well as influencing adaptive changes that could happen over time. Specifically, we asked the following research questions:

1. How can the Arts for the Blues intervention be scaled up for integration within healthcare and cultural organisations to tackle depression and improve wellbeing in communities across the North West of England?

2. What contribution can our in-depth investigation of our scaling up activities relating to Arts for the Blues make to developing a new strategy on scaling up place-based arts initiatives that support mental health and wellbeing?

We defined 'place-based arts initiatives that support mental health and wellbeing' as:

a. needing to have a particular starting point in a service, community organisation or neighbourhood and

b.  involving an intentional aim to improve mental health and wellbeing.

   Examples of these may be:

- arts and health projects that target mental health and/or wellbeing outcomes facilitated by artists;

- therapeutic uses of the arts delivered by therapists/counsellors;

- any form of arts therapies delivered by qualified arts therapists.

## Methodology

To answer the research questions and enable regional changes, we drew on the principles of realist evaluation methodology [41–44] for developing and refining a set of theoretical assumptions (the project's 'programme theory'). Specifically, we articulated four pragmatic programme theories (see **Table 1**) that could be refined in relation to scaling-up and embedding an arts-based intervention, thereby gaining learning that could be shared for future research and implementation practice. These theoretical assumptions conceptually framed how the Arts for the Blues initiative could be scaled up, under what circumstances, and how potential barriers could be pro-actively addressed. In realist evaluation terminology this referred to articulating and evaluating configurations of (a) 'contexts', which, in our study, involved mainly healthcare and cultural organisations, and (b) the 'mechanisms' for and against achieving the desired 'outcome'. In our case, the desired outcome was the integration of Arts for the Blues in regular practice within the contexts of healthcare and cultural organisations, whilst the mechanisms were the unknown factors we set out to discover.

(i) <u>If a complex intervention is simplified for initial use by practitioners with different or less experience, this will improve the spread and adoption of the innovation in the healthcare and cultural sectors</u>

   Although arts therapies are complex interventions with an integrative character, drawing on humanistic, psychodynamic, developmental, behavioural and artistic practices [45], Arts for the Blues offers a manualised psychotherapeutic group intervention that is intended to be much simpler [20]. As explained above, it was developed in response to the need to offer a wider choice in local mental health services; bringing more creative/artistic activities into the NHS [15], articulating safe ways in which to engage with the arts through the presence of appropriate framing, offering a structured use of key ingredients for psychological change and providing an informed awareness of the stages of group development.

   The model was designed primarily as a form of psychotherapy depending on the needs of the participants, the skills and qualifications of the facilitators [20], but we also explored how it

**Table 1.  Theoretical assumptions refined in the realist evaluation.**

1. If a complex intervention is simplified for initial use by practitioners with different or less experience, this will improve the spread and adoption of the innovation in the healthcare and cultural sectors.

2. Understanding the value of an intervention will support its adoption at a local level.

3. If the innovation responds to current calls for reform, it has more chances of becoming a useful addition to regular provision.

4. Engaging in vertical and horizontal activities can allow an innovation to gain influence, reach a wide user-base and advance its chances for adoption.

can be used flexibly as an arts project with a therapeutic intent within community organisations (Thurston et al., in preparation). It was therefore possible that the intervention could be delivered as a single or a multi-modal creative psychotherapeutic intervention or as a music, dance, drama, visual arts, creative writing or multi-modal arts project with therapeutic intent (more on evaluation results from the training of therapists/counsellors, artists and arts therapists, on Karkou et al., in preparation).

All these characteristics of Arts for the Blues were regarded as simplifying the complexity of arts therapies and contributing to an easier transfer, spread and adoption, an assumption we set out to refine during the study.

(ii) <u>Understanding the value of an intervention will support its adoption at a local level</u>

Our second assumption was that for organisations to be ready to adopt an arts initiative that supported mental health and wellbeing, they would need information about the initiative and relevant staff training. Our training programme on the Arts for the Blues model was developed and delivered prior to this study at two MSc programmes in Counselling and Psychotherapy at UK universities and with talking and arts therapists in the North West of England, the UK and internationally. It covered two days and involved the development of the model, its key ingredients and groupwork stages and processes. It was delivered both in person and online and involved presentations, videos and experiential small group work. By the time we began this study we had already trained over 80 qualified and trainee psychotherapists in the region, creating some of the first opportunities to spread and adopt this intervention to a wider range of clinical settings and groups. However, it became clear that our training had to involve not only artists interested in participatory arts-making with vulnerable groups and collaborative models but also frontline staff from targeted organisations and sectors in order to mobilise adoption of the model in these settings.

At the same time, we were aware that there would be differences between settings. For this reason, we found it important to investigate the views of stakeholders associated with different organisations and sectors on what would enable them to adopt Arts for the Blues as part of the routine delivery of their services.

(iii) <u>If the innovation responds to current calls for reform, it has more chances of becoming a useful addition to regular provision</u>

We considered different health and cultural reforms and related policies for their direct impact on these sectors. The Integrated Care Services reform (ICS) of 2021 [46] and developments in social prescribing [47] are two areas of focus. For example, setting up the new Integrated Care Systems and associated Integrated Care Boards created an important shift in bringing together healthcare services and community organisations. Social prescribing brought to the foreground the contribution the arts can make to supporting people's health in holistic ways, while personalised care shifted control and power from the expert to the service user [48]. Similarly, the new NICE guideline for depression [49] acknowledges that one intervention does not fit everyone, expanding on a range of therapies for depression as a first line of treatment. Initiatives at a national level such as the foundation of the National Centre for Creative Health and the Culture, Health and Wellbeing Network supported by the All-Party Parliamentary Group on Arts, Health and Wellbeing, are all driving the agenda towards the inclusion of the arts in national policies.

In terms of more general changes at a national policy level, this innovation responds to the NHS focus on investment in mental health and addressing health inequalities in line with the NHS Long Term Plan [50], bridging the gap between primary and secondary care services.

This innovation also responds to the National Institute for Health and Care Research (NIHR) list of research priorities: tackling depression ranks within the top ten [51].

(iv) Engaging in vertical and horizontal activities can allow an innovation to gain influence, reach a wide user base and advance its chances for adoption

Influenced by the WHO [52] strategy for scaling up projects we also assumed that there will be vertical and horizontal activities which will engage people with diverse roles within the wider ecosystem from frontline staff to local, national and international leads and across different sectors. The WHO [52] strategy argues that engaging with policy and national initiatives (vertical scaling up) and replicating the innovation in new areas and with new client populations (horizontal scaling up) are two important steps to gain influence and widen the user base. We therefore assumed these activities would act as enabling factors for the scalability of Arts for the Blues and any other place-based arts initiatives that support mental health and wellbeing.

## Sample

According to the WHO [52] strategy for scaling up projects, there is a need to engage with key decision-makers in local services and organisations as well as to reach a wide range of frontline practitioners. We therefore considered key stakeholders who could support the inclusion of Arts for the Blues in their services and/or held other key roles in the region that could support the scaling up of this initiative. We focused on the leaders of health and cultural organisations, mental health charities and local authorities (See **Table 2**).

We also utilised snowball sampling by asking those who agreed to participate in the study to disseminate details about the project to their staff in order to bring together frontline workers and other key professionals working with vulnerable individuals from different sectors.

To secure the value of the work for service users, throughout the process we worked closely with our PPIE group: most of them were mental health service users who had attended Arts for the Blues groups in the past and varied in age, gender, ability/disability and socio-economic, cultural and educational backgrounds. Their contribution included expanding the list of organisations we engaged, commenting on the material we produced in preparation for data collection and being enthusiastic advocates of the Arts for the Blues intervention and the use of creative methods in therapy during the data collection process. They also commented on the key findings of the study presented as an easy read version (https://www.edgehill.ac.uk/wp-content/uploads/documents/Strategy-of-scaling-up-arts-projects-with-therapeutic-impact-1-EASY-READ.pdf). Their views were included in the film we produced that offered an audiovisual summary of the project (https://artsfortheblues.com/post-2/).

**Table 2. People invited to attend stakeholders' events.**

From healthcare settings we invited:
- Leads of psychological services delivered by NHS trusts in the region.
- Leads from NHS Innovation North West Coast.
- Leads from NHS Research and Innovation.
- Leads of the newly founded regional Integrated Care Systems (ICS).

From cultural organisations we invited:
- CEOs of cultural organisations from the region, paying attention to covering representation from different art forms.
- Leads of regional partnerships in culture, health and education.

We invited community organisations including:
- CEOs of some of the largest mental health charities in the region.
- social prescribers and social prescribing leads.

Finally, we invited representatives from local authorities:
- Leads in public health, culture and related roles.

## Methods

Two stakeholders' events (one in-person and one online) and four training days for frontline staff (two in-person and two online) took place from 16/06/2022 to 20/01/2023. Participants were invited to engage in focus groups at the end of the stakeholders' events and respond to a set of open-ended questions at the end of the training days.

All of the questions were intended to test the theoretical assumptions of the study. They were informed by the integrated-Promoting Action on Research Implementation in Health Services (i-PARIHS) framework [53, 54] which is a framework extensively used in implementing research findings in health contexts [55].

Examples of the questions asked included:

Innovation

- How clear and accessible is the Arts for the Blues model?

- Who is likely to benefit from the Arts for the Blues model?

- In what ways is the evidence for Arts for the Blues convincing?

  Inner context

- What is your experience of change in your organisation?

- What is the understanding of the arts within your organisation?

- How can the model be integrated into practice and service user delivery?

  Outer context

- How do you think the Arts for the Blues model aligns with recent policy?

- In your organisation what are the opportunities for using the Arts for the Blues model?

- What would help you to work with professionals in other sectors to deliver Arts for the Blues on a wider scale?

The focus groups (9 groups in total) were either audio-taped and transcribed manually (in-person stakeholders' event) or were recorded, transcribed using the transcription function on 'MSTeams' and checked for accuracy (online stakeholders' event). At the end of each training day, participants were asked to either hand-write answers to questions on large pieces of paper (in-person training), which were subsequently transcribed or to type their responses to questions on a 'padlet' (online training).

## Analysis

We used thematic analysis [56] to generate themes which were then considered for their relevance to the theoretical assumptions of the study. We also conducted a higher order synthesis of findings which was informed by the i-PARIHS framework [53, 54].

## Ethical approval

Ethical approval was obtained by Edge Hill University and Health Research Authority (IRAS ID: 314064).

## Findings

During the study we worked directly with 225 managers and frontline workers by engaging them in stakeholders' events and training days. Amongst them, 43 attended the stakeholders' events (see **Table 3**), while 182 received training (**Table 4**).

**Table 3. Participants in stakeholders' events.**

| Type of organisation | Managers | Researchers | Practitioners | Total |
|---|---|---|---|---|
| Healthcare settings | 9 | 6 | 3 | 18 |
| Cultural organisations | 12 | 0 | 3 | 15 |
| Community organisations | 7 | 0 | 0 | 7 |
| Local authorities | 3 | 0 | 0 | 3 |
| Total | 31 | 6 | 6 | 43 |

As **Table 3** shows, 31 of the 43 participants in the stakeholders' events held managerial positions in healthcare settings, cultural and community organisations and local authorities. Some researchers and practitioners from the invited organisations also came along with their managers, engaged in discussions around the Arts for the Blues model and participated in the focus groups during the two stakeholders' events.

From the 182 people who attended the training days (**Table 4**), 143 were practitioners with different qualifications such as therapists (with qualifications in psychology, counselling or psychotherapy, mental health nursing), artists (with backgrounds in dance, music, drama, creative writing, visual arts), arts therapists (with training in dramatherapy, music therapy, dance movement psychotherapy, art psychotherapy) and other professionals (e.g. allied health professionals, nurses, volunteers). In most cases, the participants had relevant qualifications in a specific subject but there were 23 amongst them who were students on subjects related to therapy (e.g. psychology or counselling), the arts (e.g. dance, creative writing, music) or arts therapies (e.g. music therapy, dramatherapy and dance movement psychotherapy). In some cases, managers of organisations also received the training offered and responded to the specific evaluation questions asked at the end of each of the two training days.

Further demographic information relating to those attending the training were not captured, but the programme was advertised through the organisations participating in the stakeholders' events, established training programmes that the research teams were involved in, and professional associations in the UK and selected countries that had expressed interest in receiving the training, namely Czech Republic, Estonia, Latvia, Lithuania, Israel and Taiwan.

## Challenges

Participants in the study named some key challenges in scaling up place-based arts initiatives that support mental health and wellbeing.

For example, the narrow priorities of the ***policies*** they had to implement in their organisations were mentioned, restricting them from investing time and energy in new interventions, even if there was willingness in the organisation to do so. As one of the participants from a local council put it:

**Table 4. Participants in training days.**

| Type of profession | Managers | Researchers | Practitioners | | Total |
|---|---|---|---|---|---|
| | | | qualified | students | |
| Therapists | 3 | 4 | 50 | 14 | 71 |
| Artists | 6 | 1 | 30 | 6 | 43 |
| Arts therapists | 0 | 0 | 56 | 3 | 59 |
| Other professions | 2 | 0 | 7 | 0 | 9 |
| Total | 11 | 5 | 143 | 23 | 182 |

"*The current sort of policy situation is very focused on those core bits and my gut instinct is this will be seen as a nice thing to do rather than an essential to do, and therefore introducing it this time would be a challenge*" (3.COUNCIL.A—Manager)

In i-PARIHS terms [53] the outer context appeared to impact on the inner context of the operation of organisations, creating barriers in integrating arts initiatives like Arts for the Blues.

Associated with the restricted polices, guidelines and standards that organisations are required to meet, was the limited ***funding*** available in both community and health sectors, raising another important 'inner context' barrier in i-PARIHS terms [53]. However, unlike in cultural organisations, some service managers in the NHS were aware that there were pockets of funding in different services for different things, which could be possibly used for arts initiatives like Arts for the Blues:

"*there are quite a few organisations within the NHS which do have the money and are interested in innovative approaches to problems*" (1.NHS.A—Manager)

However, navigating funding opportunities as well as ***hierarchical structures*** and associated approvals was not an easy task for some organisations, the NHS in particular. Hierarchies not only challenged the process and time of approval but also, as discussed in one of our focus groups, created systematic disempowerment that did not allow people to make decisions on what they thought was a useful intervention for their clients. For example, one of the social prescribers participating in the study argued that "there's a reticence to own decision making . . ." (1.Social Prescribing.A—Manager) acting as a barrier to offering the best service possible.

Concerns about service users' ***risk*** to themselves or others added further disempowerment for organisations that did not offer arts activities on a routine basis. The same social prescriber as above suggested that these concerns were stopping the health sector from actively engaging with arts initiatives:

"*I think risk is something we use to make decisions on rather than [considering] how we manage it*" (1.Social Prescribing.A—Manager)

It is possible that such careful attention to finding solutions not only regarding risk but also regarding other practical obstacles is not readily available. Instead, it is possible that arts practitioners are seen as having a ***biased professional perspective*** and as such they overstate the value of their own work. According to a research lead and manager of one of the NHS trusts involved in the study:

"*sometimes when new therapies are introduced or even, even the existing therapies, my experience is it can be a problem being too evangelical*" (3.NHS.A—Manager)

Research evidence has been flagged up as important to commission new interventions. If this type of evidence is not available, uptake is less likely to take place. Different NHS managers appear to corroborate this:

"*In terms of kind of scaling up and implementation for new interventions, which aren't fully evidence based and were commissioned for, our hands are tied*" (2.NHS.B—Manager)

"*they would be asking about the evidence base, whether new interventions are recommended within NICE guidelines*" (3.NHS.C—Manager)

Although different types of **_evidence_** are acknowledged for their value, they are not sufficient to commission services on their own:

"*personal testimony is very compelling but when it comes to commissioning services, unfortunately they do want to see those numbers and I think that could potentially be a barrier to scaling this up*" (1.NHS.A—Research)

Finally, there are arguments that the **_nature of the arts_** is such that measurement is not possible and thus engaging in large studies that establish effectiveness such as randomised controlled trials is not viable. These arguments create tensions between what is needed from healthcare services and the perception of what can be offered by the creative world, leading to limited engagement in research:

"*it always has to be measured. And sometimes some things just can't be measured*" (3.ARTS.B —Practitioner)

The limitations in the volume and type of research in place-based arts initiatives are therefore creating barriers in scaling up this work due to the initiative itself. Ways in which these challenges could be addressed were revealed as we gathered data to refine our theoretical assumptions, as described in the following sections.

## Refining our theoretical assumptions

Contributions from stakeholders and frontline staff allowed us to refine our theoretical assumptions.

(i) <u>If a complex intervention is simplified for initial use by practitioners with different or less experience, this will improve the spread and adoption of the intervention in the healthcare and cultural sectors</u>

To refine this assumption, we presented Arts for the Blues during our focus groups and training days. To enable 'ease in transfer' [52] we produced videos illustrating each ingredient of the model and digitised examples offering creative ideas for each of these ingredients (www.artsforthebluespractice.co.uk).

We also presented results from our extensive engagement and consultation with lived experience experts, professional experts and results from empirical research with different client groups to explore whether this model of work could be appropriate:

- Not only for people with depression but also for other groups and individuals such as children with emotional and behavioural difficulties, adults with diverse mental health concerns and older people with or without dementia;

- Not only to be facilitated by arts therapists but also by other kinds of therapists as well as artists;

- Not only facilitated by professionals but also used by individuals on their own.

When we asked participants to consider whether Arts for the Blues as an arts initiative that supported mental health and wellbeing could be easily spread and adopted, we found several interesting results summarised in **Fig 1**. This offered information about what in the i-PARIHS framework is referred to as the 'innovation' construct [53].

Overall, and despite the diverse backgrounds of the study participants, they argued that the presented model was **_adaptable_** and **_clear_**. For example, participants liked: "the concepts and

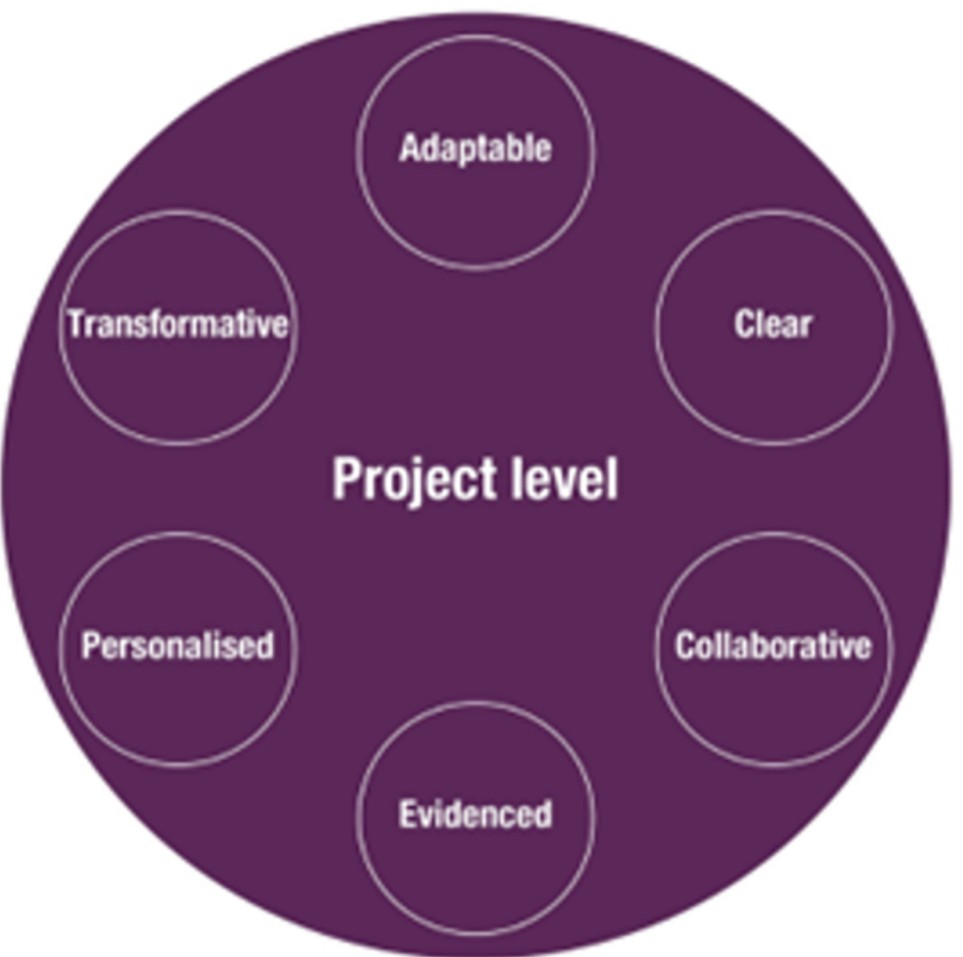

**Fig 1. Project level: Features of scalable arts initiatives such as Arts for the Blues.**

the flexibility of the model" (Frontline staff, online training day 1) and claimed that the model had a "very clear structure, leaving space for your own input, improvisation and interpretation" (Frontline staff, in-person training, day 1). Adaptability and clarity were therefore seen as two important and useful components of the specific model. It was also suggested that this was not simply an intervention relevant for people with a diagnosis of depression, but it could be potentially relevant for diverse groups of people:

> "*It doesn't have to be just a group of people who attract a particular diagnosis; it could just be a mixed group*" (1.NHS.C—Manager)

The Arts for the Blues model was seen as building good relationships and actively supporting people to engage. Its creative components were highly appreciated for achieving just this, i.e. **_collaborations_** with others and opportunities to connect:

> "*[There is a need for] methods which help to connect with others*" (Frontline staff, in-person training, day 1)

In addition, its potential to support people's empowerment and growth was also referred to.

"*when you take something therapeutic like that, and put it in an artistic space. I feel like those participants [. . .] have the potential to feel a real sense of empowerment from that*" (4. COUNCIL.A—Manager)

The evidence-based character of Arts for the Blues was also discussed and was considered as an important feature of this place-based arts initiative. As one of the stakeholders stated:

"*. . . there's a lot of emphasis on data. So, it would be very much [a need for] an evidence-based data-based intervention*" (3.COUNCIL.A—Manager)

It is therefore possible that the clear ***evidence-based*** character of the Arts for the Blues model offered opportunities for spread and adoption within different work environments, as one of the managers from a local authority argued:

"*The evidence [. . .] for Arts for the Blues feels really applicable, especially [. . .] as a local authority trying to quantify engagement*" (4.COUNCIL.A—Manager)

Incorporating both qualitative and quantitative evaluative methods and being informed by both of these kinds of research was also seen as important. As one of the participants argued, "there has to be a compromise or, a kind of melding of the two together that works most effectively" (3.ARTS.B–Practitioner). This 'melding' appeared to meet different definitions of evidence and thus could speak to diverse work environments.

Although stakeholders from the health sector argued that more evidence was needed in the form of large randomised controlled trials, there was an acknowledgment that some elements are difficult to capture. Client preferences and associated testimonials were therefore also valued by the different participants in our study.

Furthermore, client choice was of value not only within the context of research, but also as part of a shift in services and NHS services in particular towards ***personalised*** care. As a member of frontline staff wrote:

"*By providing another form of therapy = increased choice = better engagement*" (Frontline staff, in-person training, day 1)

Increased choice can certainly meet the needs of diverse people who could benefit from support with their mental health and wellbeing. As another member of frontline staff reflected, this does not only meet the diverse needs of individuals but also those of different types of communities: "[We need] new methods in approaching communities" (Frontline staff, in-person training, day 1). Some of these new methods could involve working with children, with minority groups, supporting the wellbeing of staff and so on.

In all cases, the ***transformative*** character of engaging in arts-making processes was important, and at the heart of the Arts for the Blues model. This was acknowledged by our diverse group of stakeholders and frontline practitioners, as one of the NHS managers eloquently put it:

"*[through engagement in the arts] people [are] finding internal fun, and interaction, and connection, and creativity. That's the attractive bit of it*" (1.NHS.C–Manager)

It therefore appeared that the scalable features of the Arts for the Blues model included its capacity to be adaptable, clear, collaborative, evidence-based, personalised and transformative

(see **Fig 1**). It is possible that other arts initiatives that support mental health and wellbeing could also be scaled up if these features were present at a project level.

(ii) <u>Understanding the value of an innovation will support its adoption at a local level</u>

Our data offered a wide picture beyond simply the understanding of the value of the arts within different organisations. Important features of the organisation that could adopt the Arts for the Blues model (or not) were also revealed (see **Fig 2**), making direct references to what the i-PARIHS construct refers to as 'inner context–organisation' [53].

In the first instance, it appeared important for an organisation to have a demonstrated ***need*** and an indication that an intervention such as Arts for the Blues could provide a way of addressing this need. As one of our stakeholders put it:

"*[. . .] you can demonstrate for example, that you have prevented somebody from visiting the GP three times that week because instead they've spent time with you*" (4.NHS.C—Manager)

Demonstrating that the intervention addressed a need of the service users becomes important. However, the need of the organisation might be of equal importance here given external pressures to deliver and concerns around the capacity of existing services to do so.

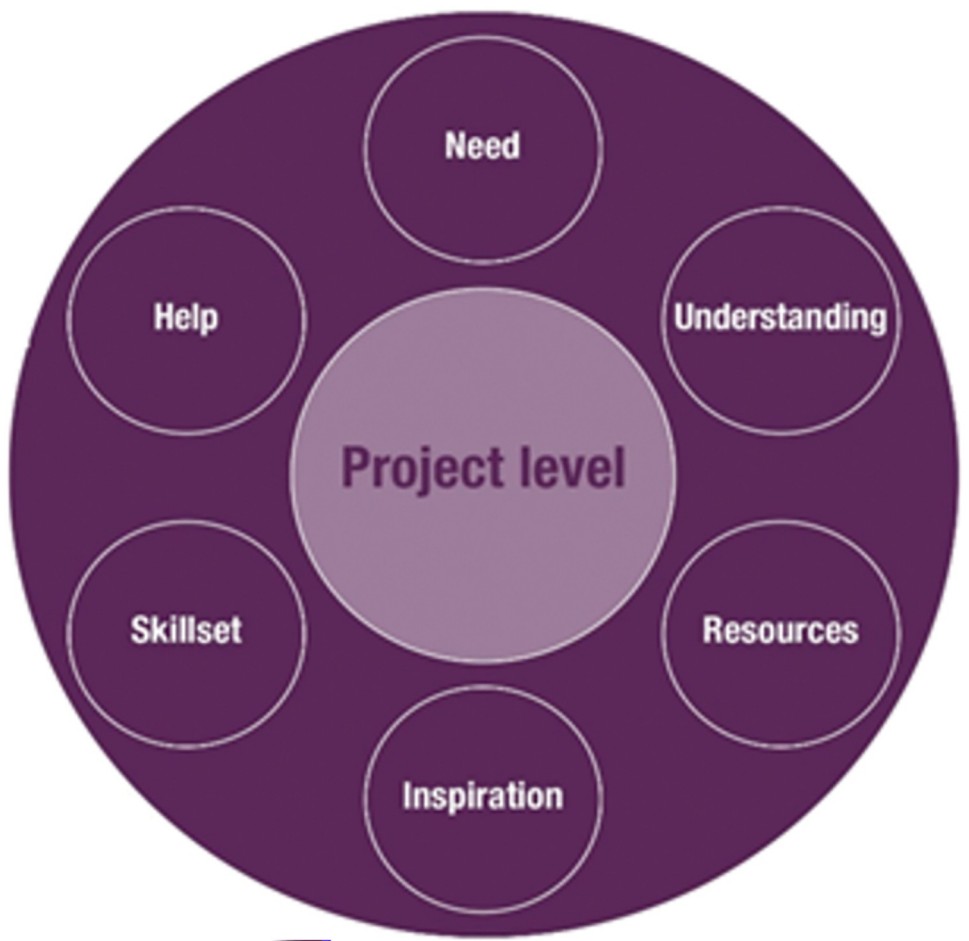

**Fig 2. Organisational level: Features of organisations that could adopt arts initiatives such as Arts for the Blues.**

As expected, study participants talked about the importance of the organisation's **_under-standing_** of the potential contribution of the arts that moves beyond leisure. As a frontline artist claimed: "arts are important, not a 'luxury extra'" (Frontline artist, in-person training day 1). Having a wider understanding of the diverse forms of impact the arts can have on one's mental health was considered as an important prerequisite for adopting arts initiatives such as Arts for the Blues. Another frontline staff member who participated in day one of the in-person training was clear that understanding of the uses of the arts also means promotion of the arts within their organisation:

"*[my organisation] understands and promotes [the arts, which goes] hand in hand [with understanding]*" (Frontline staff, in-person training, day 1)

When understanding is not there or exists only partially, very little progress can take place. One of the frontline therapists shared their experience of working in Children and Adolescent Mental Health Services (CAMHS):

"*In CAMHS my wings are clipped, unable to use creative arts. I feel that there is some understanding of the benefits but the lack of roles and interventions that are creative-based, undermines the profession and you get a sense people think it was phased out as [if] it was not effective. The emphasis is on CBT and more technical rather than relational therapies.*" (Frontline therapist, online, day 1).

The difference between these two accounts is telling of the importance of understanding as an important feature at an organisational level.

**_Resources_** to make this happen were also important. One of the NHS managers in the study suggested that through introducing an arts initiative

"*You've provided an alternative and you've saved money somewhere else in the system*" (4. NHS.C—Manager)

This was proposed by this manager as an important argument to put forward that can address some of the challenges relating to the lack of funds presented above. By contrast, in the third sector, which is often even more underfunded than the NHS, committed managers in introducing initiatives such as Arts for the Blues were willing to find solutions:

"*There's never any new funding for these types of initiatives. We just have to make it work*" (2. CHARITY.B—Manager).

The commitment of managers as budget-holders to introduce arts-based initiatives is therefore important, overcoming limited resources.

Furthermore, for making things happen, **_inspiration_** did appear to be important. In particular, the role of local champions was seen as necessary for keeping an active interest and supporting action: "innovations require a product champion" (3.NHS.A—Manager). This is a motivating factor that comes from within the organisation and acts a catalyst to action and change.

Reflecting on the training offered, people responded with very positive comments. Not only were there improvements in knowledge and skills, but there was also a boost to their confidence:

"*The training has enabled the clinical lead and a counsellor to feel more confident in presenting and working with creative material*" (Frontline therapist, in-person training day 2).

This was particularly relevant for practitioners who were not directly involved in using the arts therapeutically e.g., talking therapists or artists. Bringing therapists and artists together was therefore of clear **_help_**, and was valued by a wide range of participants in our study. As one of the participants reflected:

"*[Working with people from different backgrounds was] enriching, valuable, expanding*" (Frontline therapist, online training, day 1)

Also, it was helpful to work not only with individuals but also with different organisations. As one of the frontline therapists commented, help from different individuals and organisations can benefit service users:

"*. . . we can partner with the cultural / arts sector to improve patient experience*" (Frontline therapist, in-person training, day 1)

In summary, it became clear that at an organisational level, a place-based arts initiative such as Arts for the Blues could be adopted more easily if the organisation had the following elements: a relevant need, an understanding of the value of the arts in supporting mental health and wellbeing, appropriate resources, inspiration held by a local champion and help from individuals and/or organisations (see **Fig 2**).

(iii) <u>If the innovation responds to current calls for reform, it has more chances of becoming a useful addition to regular provision</u>

As part of engaging with calls for reform in mental health services, we considered how the Arts for the Blues model would be included as an additional option in NHS Talking Therapies and NHS-led community services. We identified gaps in provision and argued that Arts for the Blues can address these gaps.

Arts for the Blues was offered in different contexts and modified versions were delivered to suit the needs of the settings and the clients. We also aimed to address specific calls such as creative activities for self-care during the Covid-19 pandemic or arts workshops for the wellbeing of NHS staff after the pandemic. Furthermore, we adapted the model to institutional standards, changing the length and number of sessions as well as adding relevant outcome measures.

In the community and with cultural organisations we developed performances that contributed towards building public awareness, supporting the role of the arts in improving mental health and wellbeing and speaking to calls to use the arts for public health messaging.

Finally, it was important that we remained connected with important cultural and health developments at a regional and national level by contributing to relevant consultations, guidelines and governmental briefings including the NICE guidelines for depression, the government's inquiry in relation to prevention in health and social care as well as briefings and reports to the WHO.

The participants in our study also had strong views on how Arts for the Blues and similar arts initiatives could respond to calls for reform, as **Fig 3** shows. They therefore made a direct reference to what the i-PARIHS framework referred to as the 'outer context' [53].

In the first place, recent **_attitude shifts_** on how the arts were perceived needed to be acknowledged. Managers from the NHS were able to see the relevance of the arts for people with a wide variety of backgrounds as argued by one of the NHS managers ("[the arts] are probably more accessible to people from a wider variety [of backgrounds]", 1.NHS.B—Manager). These shifts were at the cusp of change not only regarding attitudes towards the arts but

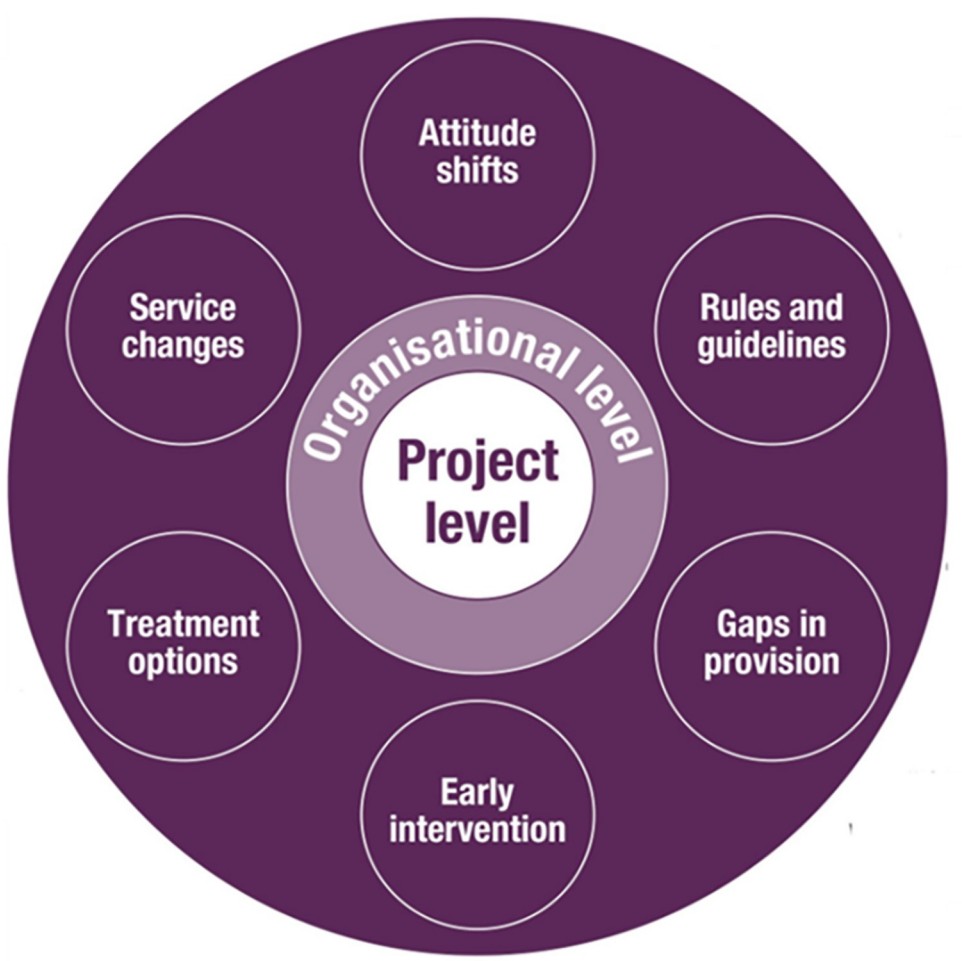

**Fig 3. Policy level: Factors supportive of arts initiatives such as Arts for the Blues.**

also regarding ways in which talking therapies were delivered: "How we offer talking therapies in [city] is looking at changing" (2.CHARITY.A—Manager).

 *Rules and guidelines* also shaped the way the work could take place. Being aware of required standards was proposed as an opportunity to introduce arts initiatives such as Arts for the Blues. One of the NHS stakeholders suggested that

> "...*there's an opportunity to say [...] we've got that menu of options [...] it then helps to achieve those standards and [...] link it to some of those targets"* (1.NHS.A—Research).

Adopting the organisational rules, guidelines and thus standards, can enable the organisation to meet its targets and as such, adoption of the arts initiative becomes much easier.

 Another stakeholder from an arts organisation argued that arts initiatives such as Arts for the Blues can fill *gaps in provision*: "You know that the sheer level of waiting times and waiting lists [...] And the NHS [...] is unable to fulfil it" (4.ARTS.C—Manager). Another suggested clever ways forward: "[...] it's always easier to innovate if you present it as a solution to acknowledged problems" (1.NHS.C—Manager).

 *Early intervention* as part of personalised care and social prescribing was seen as playing an important role in enabling arts initiatives to spread. One of the social prescribers present at our study clarified:

*"the development of personalisation as a key area within health and social prescribing within that, I think there's an alignment there"* (1.SOCIAL PRESCRIBING.A—Manager).

The capacity of arts initiatives to offer personalised rather than prescriptive care and do this early on and prior to mental health problems arising appears to be an advantage that can be used to promote scaling up activities.

Placing the arts and place-based initiatives in the local ecosystem was important. A local authority representative suggested that:

*". . .it is about being part of a diverse sort of local ecosystem and [the arts are] one of the key bits"* (3.COUNCIL.A—Manager)

Similarly, NHS staff suggested that the arts can be used in diverse ways not only for prevention but also as ***treatment options***. NHS managers themselves highlighted the need for more choice in mental health services: "I think [available primary care mental health services] don't fit everybody. . ." (4.NHS.C—Manager). Another reflected that:

"*[services can be] about approaches, which [. . .] are non-medical [. . .] particularly creative approaches can be really life enhancing*" (2.NHS.B—Manager).

One of the NHS managers described their vision of how this could resemble a system surrounding the person in need:

"*We just want a system that kind of surrounds the person and meets their needs at whatever spectrum of need they've got really. And this would kind of fill some of that; it would be fluid really*" (4.NHS.C—Manager).

Finally, at a policy level, NHS managers suggested that ***service changes*** are currently providing opportunities which were not there before. One of them explained this:

"*because of ICSs, because of the move from 200 PCT and CCGs to 46 ICSs [. . .] there's never been a better opportunity in the last 20 years to be innovative and align what we do with population need*" (1.NHS.C—Manager)

In summary, at a policy level we found that place-based arts initiatives such as Arts for the Blues needed to consider the following elements/factors: attitude shifts, rules and guidelines, gaps in provision, early intervention, treatment options and service changes; all potentially important factors to supporting scalability at this level (see **Fig 3**).

(iv) <u>Engaging in vertical and horizontal activities can allow an innovation to gain influence, reach a wide user base and advance its chances for adoption</u>

Following the WHO [52] strategy we engaged with both policy and national initiatives (vertical scaling up) which included talking and sharing information with NHS, Research and Innovation and local authority leads in Liverpool and Manchester, directors of national centres (e.g. National Centre for Creative Health and Culture, Health and Wellbeing Network), and professional associations (e.g. UKCP, BACP, Arts therapies associations).

We also engaged leads of international initiatives on Arts for Social Justice, Arts and Health and Arts Therapies, e.g., UNESCO Ambassador of Arts for Peace, the WHO leads on Arts and Health and the International Creative Arts Therapies Research Alliance.

Horizontal activities, defined by the WHO [52] strategy as an opportunity to replicate the innovation in new areas and with new client populations, involved working with people (both service users and staff) attending cultural organisations, hospitals/healthcare settings, schools, charities and other community organisations. We also achieved geographical spread by engaging with organisations in Greater Manchester, Merseyside and Lancashire.

Our horizontal activities also involved preparing feasibility studies with a range of settings for minority groups in the UK (including the NHS) and discussing Arts for the Blues projects in countries such as the Czech Republic, the Baltic countries, Israel, Pakistan, Malaysia, India, the Caribbean (Barbados, and Trinidad and Tobago) and Colombia.

## Vertical scalability

Regarding vertical scalability, participants in our study talked about both bottom up and top-down approaches (see **Fig 4**).

In some cases, they argued that their approach was primarily working with local assets:

"*. . . it's a very bottom up but asset-based way of working. We sort of work with them*" (3. COUNCIL.A—Manager)

Others, mainly smaller community organisations, offered a very ***bottom up*** approach to innovation:

"*[We might have a] therapist or a couple of therapists within the team who are champions for a new model [. . .]*" (2.CHARITY.B—Manager)

The same person explained this point further:

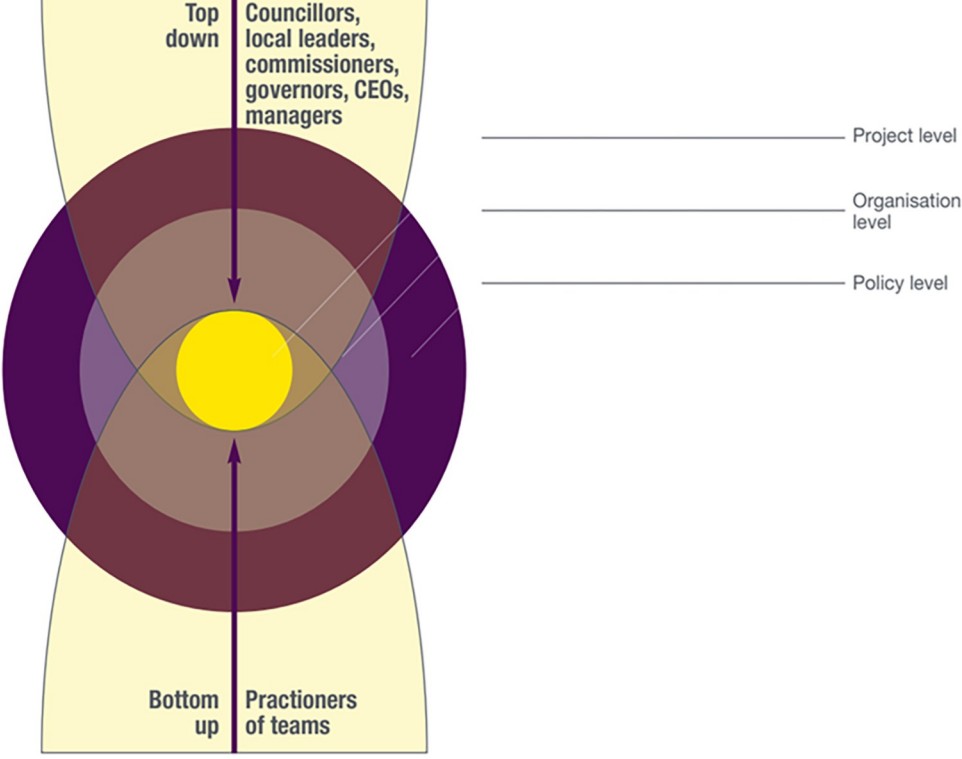

**Fig 4. Vertical scalability for arts initiatives such as Arts for the Blues.**

"*We might have a practitioner or a manager or a team who goes and does a separate training [. . .]. We might support that person and look at elements they can start to then bring in and share with the team*" (2.CHARITY.A—Manager)

In other cases, decisions from above were responsible for change in a ***top down*** manner. For example, for a local authority manager, one of the options was that new changes were introduced because of "a political decision by our county councillors." (3.COUNCIL.A—Manager). NHS managers talked about the level of governance process that they needed to go through:

"*We have to go through quite a kind of governance process now. [. . .] if it's a pilot [it needs to be] approved by the governance structures*" (3.NHS.C—Manager)

For smaller organisations, change was easier:

"[*We are] a relatively small organization where we have a CEO and three managers. So, it's fantastic. We make a decision and then we implement it*" (1.COMMUNITY ORGANISATION.A—Manager)

However, this was not as straightforward for most of the participants in our study who argued that change was hard. Still, they suggested that initiating change through the adoption of arts initiatives such as the Arts for the Blues project was important. At times this meant pushing from the bottom up, a process some found more exciting than simply responding to external drivers. However, the latter needed attention:

"*[. . .] bottom up is critical but people can't do it without permission from the top*" (2.NHS.A —Manager).

In summary, considering both bottom up and top down approaches was important in relation to vertical scalability (see **Fig 4**). Arts-related innovations could be encouraged by practitioners and teams (bottom up) and by decisions made by managers, CEOs, governors and commissioners, local leaders, councillors and, at a national level, Members of Parliament (top down). Our study participants appeared to highlight the value of both. Engaging in both types of scaling up activities was therefore of potential relevance to place-based arts initiatives that support mental health and wellbeing.

## Horizontal scalability

Finally, as **Fig 5** shows, horizontal activities were also perceived as important to enable scaling up arts initiatives such as Arts for the Blues.

Providing opportunities to widen the ***geographical accessibility*** of the work was discussed. This could take time as one of the social prescribers participating in our study suggested:

"*If the aspiration is to have [arts initiatives] across a larger geography, then it takes time in terms of both the practicalities of implementation, but also the process of approval*" (1. SOCIAL PRESCRIBING.A—Manager).

***Small pilots*** were proposed as ways in which to increase the spread of the model allowing for wider access for people with a range of backgrounds and needs:

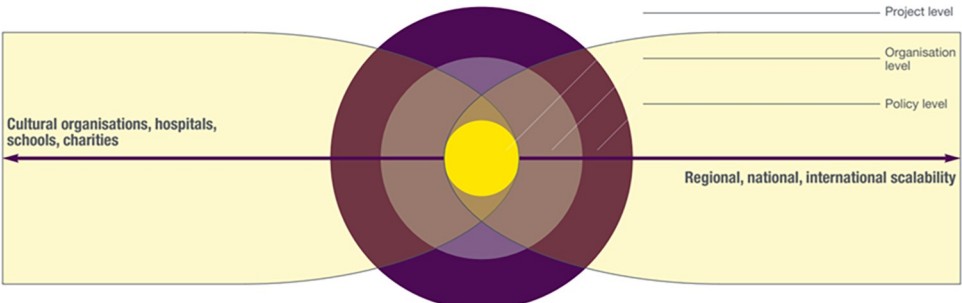

**Fig 5. Horizontal scalability for arts initiatives such as Arts for the Blues.**

"*if we were to do it as a pilot, first one I think most people would meet the criteria to come for this sort of therapy*" (1.NHS.B—Manager)

Exploring the potential use of the model with **_mixed groups_** of people e.g., with different forms of diagnoses or no diagnosis at all (as suggested by one of the NHS managers above: "It doesn't have to be just a group of people who attract a particular diagnosis; it could just be a mixed group"), appeared to indicate that the work was relevant to a wide range of people needing support with their mental health and wellbeing. Using creative interventions with different groups such as those identifying as neurodivergent was also proposed:

"*An increase in LD / ASC + mental health diagnosis would make interventions such as Arts for the Blues invaluable as an alternative approach.*" (Frontline staff, in-person training day 1).

There was an acknowledgement of the value of the work not only for primary care service users but also for staff. Concerns about the wellbeing of staff and the potential of this model to make a useful contribution came from both the health and cultural sectors: "I'm thinking of [. . .] health and well-being of our staff as well . . ." (2.NHS.B—Manager). "I suppose something that would be important for me at the moment is staff wellbeing" (2.ARTS.C—Manager).

Some of the participants went further to remind us of the role the arts have in supporting the wellbeing of **_society as a whole_**. According to one of the healthcare managers, an intervention such as Arts for the Blues "has the potential to change how we think about mental health and wellbeing." (2.NHS.A—Manager), while an arts practitioner stated that:

"i*t is kind of just, just the fact that it could be available for, for all people, all members of communities seems to be a [. . .] demand, you know of our life, for the citizens of this country*" (3. ARTS.D—Practitioner)

Within this societal benefit from engagement with the arts, a case for their relevance in engaging **_diverse communities_** was also made. It appeared important to reach out and engage with people from different ethnic minorities and thus extend therapy provision beyond current provision. According to one of the NHS mangers:

"*[Arts for the Blues] is probably more accessible to people from [. . .] a south [Asian] community, black communities, rather than it being therapy for white people and nobody else*" (1. NHS.B—Manager)

while a frontline practitioner argued that the Arts for the Blues intervention is "*urgently needed in the black community especially young adult*" (Frontline staff, in-person training, day 1).

The potential interest in this model for non-white populations triggered discussions around reaching out beyond the **_region_** and the **_country_** and thus engaging and working with **_international_** stakeholders, artists and therapists alike. Creating these connections appeared to benefit not only the users of the arts but also the practitioners themselves. One of the arts managers participating in the study suggested benefits on a research front by

"*building [shared evidence] across a number of different projects, not just within the region, but nationally as well*" (4.ARTS.A—Manager).

While one of the frontline staff who attended the training welcomed the opportunity for international exchange, a view widely shared amongst participants:

"*It is really inspiring to have such a range of practitioners, and from other countries and cultures. An important part of the inclusivity and richness of this training*" (Frontline staff, online training, day 2).

In summary, scaling up horizontally appeared to involve a geographical spread across different sectors, work environments and different individuals and communities. Small pilots across the different sectors were valued as a way into the services. Although time and effort were needed, participants in our study were clear of the imperative to make arts initiatives such as Arts for the Blues accessible for people with diverse backgrounds, thus addressing gaps in health inequalities. Horizontal spread also meant engaging with both professionals and participants beyond the region through national and international collaborations (see **Fig 5**).

## Discussion

### Research question 1

How can the Arts for the Blues intervention be scaled up for integration within healthcare and cultural organisations to tackle depression and improve wellbeing in communities across the North West of England?

To answer this question, we engaged in several activities that we expected to support the scalability of the Arts for the Blues project and similar place-based arts initiatives and asked study participants to consider a range of questions about them. This line of inquiry allowed us to consider what i-PARIHS referred to as 'innovation' [53] [54] or, in our case, as considerations at a 'project level'. For example, we presented the work we had done over the course of developing the Arts for the Blues model during all our focus groups and training days, including videos and relevant websites illustrating important aspects of the model. The project's digital presence and related guidance appeared to allow for 'easy transfer' 52 across sectors, settings and client groups.

During the course of this project, we also explored its delivery not only from arts therapists, counsellors and other talking therapists, but also by artists. Non-therapists were invited to consider the use of the model within their practice as an arts-making frame for projects with therapeutic intent. They were not encouraged to use it as therapy and/or to elicit emotional material. Further ideas included artists and therapists jointly facilitating groups, an idea particularly relevant in a region where arts therapists are not often found in mental health services, or creating joint service delivery where therapists facilitated therapy groups and artists

supported performances or exhibitions following these groups. Although the discussion is still ongoing and these ideas need further development, by bringing artists, therapists and arts therapists together to discuss the Arts for the Blues model, we hoped to kick-start rich cross pollination across these fields, maximising benefits for service users. Identifying respective skills and areas of expertise (e.g. arts-making processes for artists and capacity to work with difficult emotional material for therapists), could potentially improve current practice, tackling some serious health inequalities present in the region.

Further consideration is needed of how the arts can support one's wellbeing before mental health problems arise and before the need for input from professional expertise. Developing a toolkit of ways in which safe uses of the arts promote good mental health, can have a direct public health benefit. It is possible that the involvement of professionals with in-depth understanding of both the arts and mental/public health can support the development of this area and further advance work that is currently taking place at national [34] and international level [33] in this area.

Alongside our consideration at a project level, we also looked at the 'mechanisms' [42] responsible for scaling up place-based arts interventions such as Arts for the Blues within the 'inner context' in the form of organisations and the 'outer context' which we refer to as the policy level [53, 54]. At an organisational level, we engaged stakeholders and frontline staff in two events and four training days respectively, discussing ways in which Arts for the Blues could be scaled up in their organisations while offering opportunities to increase understanding, build resources and provide ongoing support. Further consideration is needed on how much of the original features of the model and associated evidence can be assumed when client groups, contexts and facilitators change. It is important that the value of the intervention is revisited within its new context as well as with the new client group and type of expertise of the facilitator. Examples of this can be seen in our different studies we conducted with children in schools [23, 24] and with adults in the community (Thurston et al., in preparation), all of which require exploring afresh adapted processes and intended outcomes.

At a policy level, we engaged with policy changes in the context of the NHS as well as cultural organisations and other community settings, making arguments for the potential role of Arts for the Blues in these different settings through participating in cultural and health developments and responding to consultation opportunities.

Finally, we engaged with horizontal and vertical activities as encouraged by the WHO [52] by extending our work to reach geographical spread nationally and internationally. At the same time we engaged local and national leads and connected with international initiatives on arts for social justice, arts and health and arts therapies.

Research Question 2:

What contribution can our in-depth investigation of our scaling up activities relating to Arts for the Blues make to developing a new strategy on scaling up place-based arts initiatives that support mental health and wellbeing?

The strength of our study relied on theoretically framing our work through our four theoretical assumptions. As Greenhalgh and Papoutsi [37] suggested, acquiring a clear theoretical lens was important in successfully scaling up interventions in health services. It appears that this was also relevant for arts-initiatives within and outside healthcare settings as was the case with our test project Arts for the Blues. By drawing on and articulating four pragmatic programme theories that could be refined using a realist evaluation [42] and i-PARIHS framework [53, 54], we were able to conceptualise people's opinions and behaviour in an action-oriented process that enabled the development of a new scaling-up strategy for a wider range of place-based arts initiatives (see **Table 5** for recommendations and **Fig 6** for a visual representation of the strategy as a whole).

**Table 5. Recommendations for scaling up place-based arts initiatives that support mental health and wellbeing.**

| Level | Recommendations |
|---|---|
| Project | For a place-based arts initiative that supports mental health and wellbeing to become scalable, it needs to be:<br>• **Adaptable** to address diverse mental health concerns and wellbeing needs<br>• **Clear** so it is easily understood<br>• **Collaborative** to support trusting relationships<br>• **Evidence-based** to convince different audiences<br>• **Personalised** to meet the specific needs of individuals and communities<br>• **Transformative** to energise and support change<br>**A**daptable, **C**lear, **C**ollaborative, **E**vidence-based, **P**ersonalised and **T**ransformative = **ACCEPT**. |
| | Organisational |
| For a place-based initiative that supports mental health and wellbeing to be integrated in a service, the organisation will have to have:<br>• **Need** for the particular contribution of the arts<br>• | **Understanding** of the benefits of the arts<br>• **Resources** that can support new initiatives<br>• **Inspiration** to make things happen<br>• **Skillset** to offer arts interventions that are creative and safe<br>• **Help** from other professionals and organisations<br>**N**eed, **U**nderstanding, **R**esources, **I**nspiration, **S**killset and **H**elp = **NoURISH**. |
| Policy | For a place-based arts initiative that supports mental health and wellbeing to become scalable, it needs to consider:<br>• **Attitude shifts** on how the arts are perceived<br>• **Rules and guidelines** that govern services<br>• **Gaps in provision** that need to be filled<br>• **Early intervention** options such as social prescribing<br>• **Treatment options** for those who are more vulnerable<br>• **Service changes** and opportunities that come from these changes<br>**A**ttitude shifts, **R**ules and guidelines, **G**aps in provision, **E**arly intervention options, **T**reatment options, **S**ervice changes = **tARGETS**. |
| Vertical | For a place-based arts initiative that supports mental health and wellbeing to be scaled up vertically, it is important that:<br>• Individuals and teams act as champions to encourage a bottom-up approach.<br>• Managers of services and Chief Executive Officers of organisations, local councillors, local leaders with pastoral roles, commissioners in the new Integrated Care Boards, Health governors, Members of Parliament with health or culture portfolios need to be onboard to support a top-down approach. |
| Horizontal | For a place-based arts initiative that supports mental health and wellbeing to be scaled up horizontally, it is important that:<br>• The intervention is implemented in different settings, regionally, nationally and internationally. |

For example, we assumed that if we were able to simplify an arts-based project such as Arts for the Blues, drawing heavily on complex interventions such as arts therapies and other evidence-based interventions, we would be able to support the spread and adoption of this simplified arts initiative. Our challenge, whilst doing this, was to simplify the work without becoming a prescriptive intervention, allowing for the huge diversity of practitioners with different skillsets to regard this as a useful structure of how to use the arts in their own practice with different client groups in different contexts [20]. While we engaged with stakeholders and frontline staff, other features of a scalable intervention were also highlighted such as the degree of evidence of effectiveness, its relational frame, its personalised character [48] linking with personalised shifts in care and pluralistic theoretical underpinnings [19] and ultimately, its capacity to offer transformative experiences through the key mechanisms of psychological change [20]. Considering all these features in a place-based arts initiative that supports mental health and wellbeing is important when one is devising scalable projects.

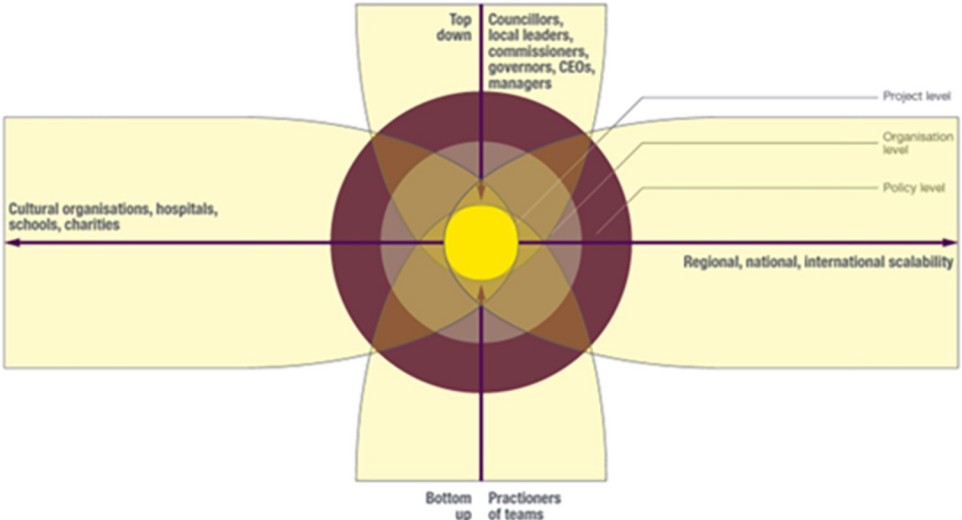

**Fig 6. Strategy of how to scale up place-based arts initiatives that support mental health and wellbeing.**

Our second theoretical premise included our assumption that for organisations to adopt these types of initiatives they had to have an identified need and a clear understanding of the potential contribution the arts can make to one's mental health and wellbeing. As AHSN [36] reminds us, organisations can be complex, requiring equally complex and often multi-layered approaches to be able to integrate new initiatives. This may become yet more complex if the organisation is lacking an understanding of what the arts can potentially contribute. This is often closely linked with allocating resources, investing in developing relevant skillsets and committing to overall capacity building; all important considerations, as we have found, for an organisation to adopt this type of initiative and equally important in the relevant literature [52]. The economic benefits of engaging in the arts may also need to be considered further adding to emerging research [29] and made available for use within different organisations. Inspiration through, for example, champions and commitment from inside the organisation alongside support from outside through input from individuals with expertise and/or collaborations with other organisations, offered a further and more nuanced understanding of the important requirements for scaling up arts initiatives at an organisational level [36].

Our third theoretical assumption related to the need for initiatives to respond to current calls for reform. We also found that not only was it important to be aware of relevant policies, rules, guidelines and standards associated with these policies, but also of attitude changes in how the arts are perceived and gaps in provision where the arts can act as useful additions. We also noted the awareness stakeholders had of the role of the arts as early intervention in the form of social prescribing [47], for example, changes in treatment options such as shifts to community and integrated care [46] and changes in how services are organised such as the new NHS Talking Therapies [6]. Our initial assumption was therefore further nuanced beyond the i-PARIHS [53, 54], highlighting specific areas of pressure from outer contexts that are particularly relevant to arts initiatives that support mental health and wellbeing.

Finally, we hypothesised that, as recommended by WHO [52], if we engage in vertical and horizontal activities, arts initiatives such as Arts for the Blues will gain influence, reach a wider user base and advance its chances for adoption. It was apparent that engaging in such activities allowed us to achieve successful engagement with diverse services, inspire teams to engage, and support adoption of the intervention. It also became clear that focusing on one client

group such as people with depression was a limiting factor as such arts interventions could be potentially useful for a wider range of different client groups including people with diverse mental health and wellbeing needs, and including both service users and staff.

## Revisiting challenges

While reflecting on the challenges that were named during our study, it appears that there were several ways in which these challenges could be addressed. For example, study participants argued that the current policies were restrictive, limiting them from adopting new arts initiatives such as Arts for the Blues. However, as personalised care, user choice, social prescribing and other holistic approaches to care are gradually entering governmental policies [47–49], the opportunities to introduce new arts interventions are now more relevant than ever. As healthcare managers also suggested, the new ICS [46] provides important opportunities to reconsider what care is offered, how it is delivered and by whom, creating promising opportunities for the future.

One implication of some of the above changes in policy is the development of funding opportunities for new areas of work and new services. This suggests potential funding support for the arts in general and for place-based arts initiatives that support mental health and wellbeing in particular. Identifying funding routes to invest in creative interventions will be essential.

Associated with these opportunities is the challenge of changing ingrained power dynamics often present in organisations such as the NHS. Change can happen when people are able to engage with what they believe to be good practice without feeling that the hierarchical structures are disabling them. Empowerment is therefore an important first step to enable change to happen. Engagement in the arts may offer this if attention to empowerment is at the heart of the intervention.

Further attention is also needed to the innovation itself. Stakeholders regarded considerations around risk as a potential barrier from referring clients to arts initiatives. Careful attention to managing risk could support decision-making processes and offer creative opportunities for novel interventions.

Another way of supporting decision-making processes is to provide robust evidence. In health contexts, evidence often consists of research evidence, client preferences and expert opinion [57]. It is possible that expert opinion from arts practitioners may be seen as potentially carrying professional bias, as one of the research leads in an NHS trust argued. Instead, client testimonies and research studies may be needed to address this perception. Doing this through support from inter-disciplinary teams may be a way of creating robust evidence, currently not readily available to policy-makers.

Although some argue that the very nature of the arts makes outcome research difficult, we have argued that an innovation that is evidence-based and well-evaluated in ways that are appropriate to its nature may offer a good prognosis for its scalability. It is also possible however, that as MacInnes et al., [58] argue, some interventions may not be scalable (or have limited scalable capacity) because they are meaningful only as place-based initiatives. They call this 'the scale and spread paradox'. As such, expansion of its use to people beyond those they initially serve and the place where it stemmed from may not be appropriate. With arts initiatives, the nature of the arts itself and their relationship with people and contexts may be such that the scale and spread paradox may be relevant. Consideration of whether this is the case or not is needed on a project-by-project basis.

Our study also generated insights which may help researchers to refine existing models used to illustrate scaling up processes. It has provided additional depth and granularity to the four domains identified by Willis et al., [35] for scaling up health care interventions. Whilst

Willis et al., [35] speak in general terms of the need for awareness, commitment, confidence and trust, our study revealed specific mechanisms which influence the success and failure of scaling up and sustaining arts initiatives that support mental health and wellbeing.

Our findings also confirm Greenhalgh and Papoutsi's [37] conclusion from systematically reviewed evidence that successful scaling up initiatives are likely to employ a mix of complexity and implementation science approaches. In particular, their emphasis on the need to pay attention to the social dynamics of programmes appears to be borne out by our findings which stress the collaborative nature of efforts to scale up place-based arts initiatives in healthcare and community sectors. The interplay between various conceptual lenses and scaling up as a pragmatic approach that addresses diverse contexts was validated in our findings and supported by stakeholders' awareness that innovations needed to be adaptable, clear and evidence based.

Moreover, our theme of increasing skillsets and resources calls attention to the need to build capacities around management, planning and implementation which may require training and professional development for small or medium sized organisations. For such organisations, dealing with large and complex structures such as the NHS may amplify existing power asymmetries in health care provision which may, in turn, influence the successful scale up of arts initiatives that support mental health and wellbeing. Since arts-based interventions usually occur as collaborative enterprises at the intersection between NHS and voluntary sector organisations, challenges around inequitable capabilities and resources may be as yet underexplored in the existing scaling up literature. Our study contributes important initial insights and provides useful points of departure for future investigations.

## Limitations

The sustainability of adopting arts initiatives that support mental health and wellbeing such as Arts for the Blues as a long-term service option is still not tested across services and over time. Furthermore, we engaged health and cultural organisations as priorities, but services for children (including schools) were not actively invited to participate in this study. Several community organisations were not included either, mainly due to limits of time and resources. Three North West local authorities were engaged, but these can also be more closely attended to in future studies to create further layering of the different sectors of local systems.

Links with national and international initiatives also need to be further consolidated, including engagement with large trials and large surveys of users that can offer solid evidence of the value of the Arts for the Blues intervention and similar interventions that offer support to specific client populations.

Although we engaged with several stakeholders and frontline staff from diverse backgrounds and sectors, the strategy is based on one arts initiative that supports mental health and wellbeing. The usability of this work for different arts initiatives needs to be tested in a second round of implementation work and be revised as needed to reflect scaling up across a wide range of place-based arts initiatives.

## Conclusions

Our study allowed us not only to identify helpful mechanisms of how to scale up specific projects but also to formulate a unique scaling up strategy for place-based arts initiatives that support mental health and wellbeing. As a result, we are able to propose actionable recommendations for the wider healthcare system and the cultural sector, focusing on opportunities that can overcome implementation barriers.

In the case of Arts for the Blues, our test project, by the end of our study there were several organisations that considered it as part of routine delivery in the UK (including two NHS

Talking Therapies services, two cultural organisations and two mental health charities). As some of the stakeholders involved in this study suggested:

> "*we've just found that the Arts of the Blues model just really spoke to us and to what we were trying to do*" (4.ARTS.C—Manager)

Another asserted:

> "*I think we are on the right track, and we might need just a further conversation on how to implement this*" (4.COMMUNITY ORGANISATION.A—Manager)"

Serious consideration of the proposed strategy is therefore needed at a local, national and international level that will allow arts initiatives such as Arts for the Blues to become scalable innovations. It appears that this is the right moment to be exploring these questions, at a time when new opportunities appear possible.

## Acknowledgments

We warmly and gratefully acknowledge the contribution of our PPIE members, all experienced in arts interventions and the Arts for the Blues model (see https://artsfortheblues.com/model/ for a short film containing their views on this specific intervention). A film relating to the focus groups and training offered as part of this project can also be found here: https://artsfortheblues.com/post-2/

## Author Contributions

**Conceptualization:** Vicky Karkou, Joanna Omylinska-Thurston, Scott Thurston, Axel Kaehne, Mark Pearson.

**Data curation:** Rebecca Clark, Emma Perris.

**Formal analysis:** Vicky Karkou, Rebecca Clark.

**Funding acquisition:** Vicky Karkou, Joanna Omylinska-Thurston, Scott Thurston.

**Investigation:** Vicky Karkou.

**Methodology:** Axel Kaehne, Mark Pearson.

**Project administration:** Emma Perris.

**Resources:** Rebecca Clark, Emma Perris.

**Supervision:** Vicky Karkou, Joanna Omylinska-Thurston.

**Validation:** Vicky Karkou.

**Visualization:** Vicky Karkou.

**Writing – original draft:** Vicky Karkou.

**Writing – review & editing:** Vicky Karkou, Joanna Omylinska-Thurston, Scott Thurston, Rebecca Clark, Axel Kaehne, Mark Pearson.

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
