## [Decision Letter · Decision Letter 0]

3 Oct 2023

PONE-D-23-11476Developing a Strategy to Scale up Place-Based Arts Initiatives that Support Mental Health and Wellbeing: A Realist Evaluation of ‘Arts for the Blues’PLOS ONE

Dear Dr. Karkou,

Thank you for submitting your manuscript to PLOS ONE. After careful consideration, we feel that it has merit but does not fully meet PLOS ONE’s publication criteria as it currently stands. Therefore, we invite you to submit a revised version of the manuscript that addresses the points raised during the review process.

We look forward to receiving your revised manuscript.

Kind regards,

Soham Bandyopadhyay

Academic Editor

PLOS ONE

Journal Requirements:

"2022-01-31 to 2023-01-30 | Grant

Arts and Humanities Research Council (Swindon, GB)

GRANT_NUMBER: AH/W007983/1"         

Reviewers' comments:

Reviewer's Responses to Questions

**Comments to the Author**

1. Is the manuscript technically sound, and do the data support the conclusions?

Reviewer #1: Partly

2. Has the statistical analysis been performed appropriately and rigorously? 

Reviewer #1: N/A

3. Have the authors made all data underlying the findings in their manuscript fully available?

Reviewer #1: No

4. Is the manuscript presented in an intelligible fashion and written in standard English?

Reviewer #1: Yes

5. Review Comments to the Author

Reviewer #1: This paper addresses the important issue of the scalability and spread of arts and health interventions. Overall, this paper proposes a plausible and relevant set of considerations to guide stakeholders, including policymakers, health care organisations, community partners, and professionals/practitioners involved in delivering arts and health programmes. It is well-written and accessible to a general audience. However, the overall argument of the paper regarding adaptation and scale-up is fundamentally weakened by the following issues:

1.  Fidelity to the original intervention and its associated evidence base is a key element of scaling up an intervention that promises and intends to achieve similar outcomes. Theoretical assumption (i) first described on P11 suggests that a simplified version of the Arts for the Blues programme delivered by different practitioners/those with less experience will support spread and adoption. However, the model programme, as described in reference 20, is based on“arts-based or creative psychotherapy […] practices delivered by a range of qualified therapists including arts and expressive psychotherapists.” Neither that paper nor the paper under review describe an adaptation that could be equivalently delivered by non-therapists or artists.

This is problematic on several points:

a. The model programme described in reference 20 contains several elements that involve the elicitation and management of emotional material that may require the guidance or support of a trained therapist. Many non-therapists lack the technical skill to practice in a way that is emotionally safe for either themselves or their participants. Most non-therapists would be reluctant to take on this responsibility. In particular, artists are quite clear that  they do not want to be therapists when working in a health and wellbeing context. In addition to the potential patient risks that concern many NHS and social prescribing stakeholders (as described in this paper, eg P15), artists themselves may suffer when confronted with their participants’ emotional distress. Also troubling is the suggestion that the intervention could be "Not only facilitated by professionals but also used by individuals on their own." (P17)

b. Many of the stakeholders cited in this paper are concerned with having a solid evidence base for adopting new interventions. The evidence for the initial programme was based on being delivered by arts therapists who are trained to provide emotional support. Since there is no discussion of the mechanisms that deliver the desired outcomes (is it the art? or the psychotherapeutic support?), we cannot assume that an adaptation delivered by non-therapists will produce the same outcomes. This ambiguity/contradiction becomes problematic in the discussion on the evidence base on PP16 and 17 and at other points in the paper.

These challenges are not adequately addressed in the later discussion.

Final approval of this paper should be contingent on a more detailed explanation of the essential elements and mechanisms of the programme’s adaptation, addressing the training and skills to required to deliver the intervention and whether non-therapists can manage complex emotional needs that may arise, and resolving the fidelity/adaptation/evidence contradictions raised above. Because some of the quotes provided as evidence do not always support the associated points, consideration should also be given to the additional items below.

Specific points or questions:

P13: are the dates for the events stated correctly here?

P13: what does ‘work closely’ with PPIE group mean? – their viewpoints are not included in this paper.

P18 after (ii) through the first paragraph on P19  – these seems like a description of methods that have already be relayed at the top of the findings section and in the methods. Similarly, the context is missing in in the first paragraphs after (iv) on P22. Vertical activities related to what?

Overall, it’s not clear how the general assertions about the historical project fit alongside data resulting from the specific stakeholder activities you describe in the methods. Is the data you’re presenting in support of your assumptions both from the years of project implementation or the stakeholder events or both? This exposition style should be made clear at the outset of your discussion of the assumptions.

P 19: under the finding of Need – this quote begs the question of whether there is any impact on GP visits from this intervention.  And more broadly – meeting whose needs? Health service utilisation targets or addressing patients’ mental health needs? .

P 19: Under the topic Understanding, the artist quote ironically reinforces the point made on P16 about having a biased professional perspective. I’m also not convinced of the point the second quote makes.

P19: The quotes under Resources are similarly unconvincing, and in direct contradiction with the resource issues raised in the Challenges section.

P19: Under Inspiration, is the training an example of something that facilitates scale-up? Is this inspiration or skills-building? Perhaps this is better placed under “Help”.

P21: The first sentence beginning “Rules and guidelines…” does not say how the project does address rules and guidelines. I’m not sure how the subsequent quote relates to the intervention meeting required standards.

P21: the first quote for the section on Early Intervention is not about that subject.

P23, near the bottom:  “Expanding the potential use” – this sentence is a statement without substantiation.

P24:  I think the quote related to Diverse communities is ambiguous and possibly subject to misinterpretation. Also, what is a ‘south western community’?

6. PLOS authors have the option to publish the peer review history of their article (what does this mean?). If published, this will include your full peer review and any attached files.

Reviewer #1: No

---

## [Author Response · Author response to Decision Letter 0]

30 Oct 2023

We uploaded the following files:

1. The rebuttal letter titled 'Response to Reviewer' 

2. A marked-up copy of your manuscript that highlights changes made to the original version titled 'Revised Manuscript with Track Changes'.

3. An unmarked version of your revised paper without tracked changes titled 'Manuscript 30 10 2023'.

The updated statement is now included in the cover letter.

The figure file is now converted in multiple PACE table files.

The dataset is now available on figshare with the following DOI: https://doi.org/10.25416/edgehill.24456367.v1

1. The manuscript meets PLOS ONE's style requirements, including those for file naming (see also above). 

2. The funders had no role in study design, data collection and analysis, decision to publish, or preparation of the manuscript. Th is is now included in the cover latter.

3. There are no restrictions on accessing data and the dataset is now made available through figshare: https://doi.org/10.25416/edgehill.24456367.v1

Response to the reviewer's comments can also be seen in the rebuttal letter but included here too.

1. Is the manuscript technically sound, and do the data support the conclusions? 

Reviewer #1: Partly 

Response: Edits made on the manuscript should address comments made 

2. Has the statistical analysis been performed appropriately and rigorously? 

Reviewer #1: N/A N/A 

3. Have the authors made all data underlying the findings in their manuscript fully available? 

Response: These are now available on Figshare, the Edge Hill’s shared repository. The DOI is now included:

https://doi.org/10.25416/edgehill.24456367.v1

4. Is the manuscript presented in an intelligible fashion and written in standard English? 

Reviewer #1: Yes

Response: We would like to thank the reviewer for acknowledging the quality of the written piece. See also additional proof reading comments made on the version with track changes.

Reviewer #1: This paper addresses the important issue of the scalability and spread of arts and health interventions. Overall, this paper proposes a plausible and relevant set of considerations to guide stakeholders, including policymakers, health care organisations, community partners, and professionals/practitioners involved in delivering arts and health programmes. It is well-written and accessible to a general audience. 

Response: Again, we appreciate that the reviewer acknowledges the importance of the topic and the value of the considerations proposed. 

1. Fidelity to the original intervention and its associated evidence base is a key element of scaling up an intervention that promises and intends to achieve similar outcomes. Theoretical assumption (i) first described on P11 suggests that a simplified version of the Arts for the Blues programme delivered by different practitioners/those with less experience will support spread and adoption. However, the model programme, as described in reference 20, is based on“arts-based or creative psychotherapy […] practices delivered by a range of qualified therapists including arts and expressive psychotherapists.” Neither that paper nor the paper under review describe an adaptation that could be equivalently delivered by non-therapists or artists.

Response: The model was indeed originally developed for therapists (arts and talking therapists) as discussed in Omylinska-Thurston et al (2020); this constituted a first level of simplification of arts psychotherapies practice for the purposes of research and scaling up of the work. 

We also explored the use of the model within a community context in the form of a therapeutic group co-facilitated between a therapist and an artist that involved running a therapeutic group that led to a public immersive performance co-designed with services users and artists. This was a project that was funded by the Arts Council (Thurston et al, in preparation). Reporting on adaptations of the model for use within this particular community context is therefore outside the scope of the paper under review. However, this project is now clearly referenced within the paper as an example of how the model can be used within this different context and with the involvement of therapists and artists.

During the scaling up study reported here, we explored if there were benefits in bringing artists, therapists and arts therapists together to discuss the model. We highlighted that there were different skills, expertise and limits of one’s practice and proposed further considerations of how artists and therapists could work together using this model, enabling cross pollination that could enrich practices and maximise access to safe uses of the arts for people who need it. This is now explicit on the paper as shown here later on.

See under ‘methodology’, in the first theoretical assumption, clarification of use of the model within a community-based group that involved both therapists and artists:

“The model was designed primarily as a form of psychotherapy depending on the needs of the participants, the skills and qualifications of the facilitators [20], but we also explored how it can be used flexibly as an arts project with a therapeutic intent within community organisations (Thurston et al in preparation).”

a. The model programme described in reference 20 contains several elements that involve the elicitation and management of emotional material that may require the guidance or support of a trained therapist. Many non-therapists lack the technical skill to practice in a way that is emotionally safe for either themselves or their participants. Most non-therapists would be reluctant to take on this responsibility. In particular, artists are quite clear that they do not want to be therapists when working in a health and wellbeing context. In addition to the potential patient risks that concern many NHS and social prescribing stakeholders (as described in this paper, eg P15), artists themselves may suffer when confronted with their participants’ emotional distress. Also troubling is the suggestion that the intervention could be "Not only facilitated by professionals but also used by individuals on their own." (P17). 

Response: This is indeed a programme developed, initially, for therapists. We do agree that artists are not trained therapeutically and as such lack the technical expertise to practise safely. We also acknowledge that talking therapists (unlike arts therapists), not trained in the arts, are in need of important enrichment in the use of creativity within their practice.

Furthermore, while we recognise and agree with these debates and issues, the scaling up proposition intended to bridge the fields of arts and health and arts therapies creating a common language, maximising on the quality of service provision within the regional socioeconomic characteristics and available resources.

In the North West of England the arts and health movement is very strong and, while practitioners working in the NHS or in the community sector often continue defining themselves as artists, the work they do is often explicitly aiming to improve people’s wellbeing. Creating opportunities for them to work safely is therefore an important public health priority that became a tacit exploration during this study. 

Regarding the use of the arts on one’s own, this is a common practice from the beginning of humanity that, during the pandemic in particular, saved people’s lives. We believe that offering guidance on how to do this safely and effectively, is an important scaling up activity that can be best generated by qualified practitioners. See additional clarification in the discussion on this topic.

The following is also added under ‘discussion’, in the first research question:

“During the course of this project, we also explored its delivery not only from arts therapists, counsellors and other talking therapists, but also by artists. Non-therapists were invited to consider the use of the model within their practice as an arts-making frame for projects with therapeutic intent. They were not encouraged to use it as therapy and/or to elicit emotional material. Further ideas included artists and therapists jointly facilitating groups, an idea particularly relevant in a region where arts therapists are not often found in mental health services, or creating joint service delivery where therapists facilitated therapy groups and artists supported performances or exhibitions following these groups. Although the discussion is still ongoing and these ideas need further development, by bringing artists, therapists and arts therapists together to discuss the Arts for the Blues model, we hoped to kick-start rich cross pollination across these fields, maximising benefits for service users. Identifying respective skills and areas of expertise (e.g. arts-making processes for artists and capacity to work with difficult emotional material for therapists), could potentially improve current practice, tackling some serious health inequalities present in the region.”

Regarding the use of the arts on one’s own, in order to make these comments explicit in the paper we added under discussion the following:

“Further consideration is needed of how the arts can support one’s wellbeing before mental health problems arise and before the need for input from professional expertise. Developing a toolkit of ways in which safe uses of the arts promote good mental health, can have a direct public health benefit. It is possible that the involvement of professionals with in-depth understanding of both the arts and mental/public health can support the development of this area and further advance work that is currently taking place at national 34 and international level 33 in this area.” 

Response: Many of the stakeholders cited in this paper are concerned with having a solid evidence base for adopting new interventions. The evidence for the initial programme was based on being delivered by arts therapists who are trained to provide emotional support. Since there is no discussion of the mechanisms that deliver the desired outcomes (is it the art? or the psychotherapeutic support?), we cannot assume that an adaptation delivered by non-therapists will produce the same outcomes. This ambiguity/contradiction becomes problematic in the discussion on the evidence base on PP16 and 17 and at other points in the paper. The evidence provided is indeed based on the use of the model as a manualised form of arts therapy, delivered by arts therapies and/or other similarly qualified practitioners. It is indeed not clear if it was delivered by other practitioners such as artists, the outcome will remain the same. However, the argument made here is not that the Arts for the Blues model, when scaled up and delivered by different practitioners, it will retain the same evidence, but that an arts project that aims to improve one’s mental health and wellbeing will need to be evidence-based and well evaluated as was the example used in this case, ie the Arts for the Blues model.

Under ‘discussion’, first research question, we added:

“Further consideration is needed on how much of the original features of the model and associated evidence can be assumed when client groups, contexts and facilitators change. It is important that the value of the intervention is revisited within its new context as well as with the new client group and type of expertise of the facilitator. Examples of this can be seen in our different studies we conducted with children in schools 23 24 and with adults in the community (Thurston et al in preparation), all of which require exploring afresh adapted processes and intended outcomes.” 

Final approval of this paper should be contingent on a more detailed explanation of the essential elements and mechanisms of the programme’s adaptation, addressing the training and skills to required to deliver the intervention and whether non-therapists can manage complex emotional needs that may arise, and resolving the fidelity/adaptation/evidence contradictions raised above. Because some of the quotes provided as evidence do not always support the associated points, consideration should also be given to the additional items below. 

Response: The adaptations for working with artists are outlined above including using the Arts for the Blues as an arts-making frame and/or working jointly with therapists.

See also changes in the discussion section where the possible uses of the model as well as the different skills of artists and therapists are named. 

In the same section, evidence is considered and the need for ongoing evaluation is highlighted.

Under ‘methodology’, a clear reference is made to the training offered which generated additional and rich material and the associated paper in preparation: 

“(more on evaluation results from the training of therapists/counsellors, artists and arts therapists, on Karkou et al in preparation).”

P13: are the dates for the events stated correctly here?

Response: This was an error that is now corrected. 

The dates of the data collection are corrected to 16/06/2022 to 20/01/2023.

P13: what does ‘work closely’ with PPIE group mean? – their viewpoints are not included in this paper.

Responsse: This is an important omission which is now rectified. Further explanations of the role of PPIE are now added. See last paragraph of the section ‘sample’:

“To secure the value of the work for service users, throughout the process we worked closely with our PPIE group: most of them were mental health service users who had attended Arts for the Blues groups in the past and varied in age, gender, ability/disability and socio-economic, cultural and educational backgrounds. Their contribution included expanding the list of organisations we engaged, commenting on the material we produced in preparation for data collection and being enthusiastic advocates of the Arts for the Blues intervention and the use of creative methods in therapy during the data collection process. They also commented on the key findings of the study presented as an easy read version (https://www.edgehill.ac.uk/wp-content/uploads/documents/Strategy-of-scaling-up-arts-projects-with-therapeutic-impact-1-EASY-READ.pdf). Their views were included in the film we produced that offered an audiovisual summary of the project (https://artsfortheblues.com/post-2/).”

P18 after (ii) through the first paragraph on P19 – these seems like a description of methods that have already be relayed at the top of the findings section and in the methods. Similarly, the context is missing in in the first paragraphs after (iv) on P22. Vertical activities related to what?

Response: These few paragraph are now deleted to avoid repetitions.

Context is added after (iv). 

We deleted the following section:

“The stakeholders’ events took place ... reached people from further afield.”

We added:

“Following the WHO 52 strategy as an opportunity to replicate the innovation in new areas and with new client populations, included...”

And

“Horizontal activities, defined by the WHO 52 strategy as an opportunity to replicate the innovation in new areas and with new client populations, involved...”

Overall, it’s not clear how the general assertions about the historical project fit alongside data resulting from the specific stakeholder activities you describe in the methods. Is the data you’re presenting in support of your assumptions both from the years of project implementation or the stakeholder events or both? This exposition style should be made clear at the outset of your discussion of the assumptions.

Response: The development of the model is clarified in several places within the paper (see page 2 of the ‘introduction’ in particular). The activities that did specifically happen within the time of this project are now further clarified under the ‘discussion’ section.

Under the discussion of the assumptions, please see edits as follows:

For example, we developed and presented the work we had done over the course of developing the Arts for the Blues model during all our focus groups and training days, including videos and relevant websites illustrating important aspects of the model. The project’s digital presence and related guidance intended to simplify its delivery appeared to allow for ‘easy transfer’ 52 across sectors and settings. Furthermore, during the course of the study, we explored its relevance for different client populations and its delivery by different type of professionals such as arts therapists, counsellors and other talking therapists, as well as artists.”

P 19: under the finding of Need – this quote begs the question of whether there is any impact on GP visits from this intervention. And more broadly – meeting whose needs? Health service utilisation targets or addressing patients’ mental health needs? .

Response: For the Arts for the Blues, there is no evidence to support any claim around a reduction of GP visits. However, we are currently generating evidence of reduction of wait times for children and young people attending CAMHS and reduction of dropout rates for adults in NHS Talking Therapies. However, since the Arts for the Blues is presented as simply an example here, presenting this evidence does not appear relevant at this point; also, this evidence is not ready to be published as yet. 

However, a reflection on the needs of the organisation is highlighted now and added to the text. 

We add here:

“Demonstrating that the intervention addressed a need of the service users becomes important. However, the need of the organisation might be of equal importance here given external pressures to deliver and concerns around the capacity of existing services to do so.”.

P 19: Under the topic Understanding, the artist quote ironically reinforces the point made on P16 about having a biased professional perspective. I’m also not convinced of the point the second quote makes.

Response: We can see the point raised here and a biased view may indeed be presented. Still, we think that an understanding of the value of the arts is an essential pre-requisite for adopting creative interventions within an organisation, more so within health environments. For this reason we kept the current quotes but in order to strengthen this section, a different experience is added from a frontline therapist working in CAMHS, where lack of understanding limits the possible inclusion of creative work within this service. This is now added to the text:

“When understanding is not there or exists only partially, very little progress can take place. One of the frontline therapists shared their experience of working in Children and Adolescent Mental Health Services (CAMHS):

“In CAMHS my wings are clipped, unable to use creative arts. I feel that there is some understanding of the benefits but the lack of roles and interventions that are creative-based, undermines the profession and you get a sense people think it was phased out as [if] it was not effective. The emphasis is on CBT and more technical rather than relational therapies.” (Frontline therapist, online, day 1). 

The difference between these two accounts is telling of the importance of understanding as an important feature at an organisational level.”

P19: The quotes under Resources are similarly unconvincing, and in direct contradiction with the resource issues raised in the Challenges section.

Response: The positive position of both NHS and charity managers cannot be under-estimated and including their position here is important. If the budget holders want to adopt creative interventions, they can make this happen. We have now added this explicitly in the text, we included the contradiction with the challenges presented earlier and added the need for health economic calculation in the discussion. 

Contradictions to the presented challenges are included here:

“This was proposed by this manager as an important argument to put forward that can address some of the challenges relating to the lack of funds presented above.”

And:

“The commitment of managers as budget-holders to introduce arts-based initiatives is therefore, important, overcoming limited resources.”

Under ’discussion’, second research question, we added: 

“The economic benefits of engaging in the arts may also need to be considered further adding to emerging research 29 and made available for use within different organisations.”

P19: Under Inspiration, is the training an example of something that facilitates scale-up? Is this inspiration or skills-building? Perhaps this is better placed under “Help”.

Response: Inspiration appears to be more important than either skills or external help. The concept of inspiration is closely linked with having people from inside the institution that are committed, ie ‘inspired’, to make things happen. The presence of Champions is also linked with this concept. We added a clarification there:

“This is a motivating factor that comes from within the organisation and acts a catalyst to action and change.”

P21: The first sentence beginning “Rules and guidelines…” does not say how the project does address rules and guidelines. I’m not sure how the subsequent quote relates to the intervention meeting required standards.

Response: This section is not about how the intervention meets required standards but how an arts initiative such as Arts for the Blues can be scaled up (see second research question). Meeting rules, guidelines and thus standards appear to be essential requirements for scaling up this type of interventions as oppositive to what often happens, they are not linked with organisational requirements for service provision.

We added:

“Adopting the organisational rules, guidelines and thus standards, can enable the organisaiotn to meet its targets and as such, adoption of the arts initiative becomes much easier.”

P21: the first quote for the section on Early Intervention is not about that subject.

Response: Early intervention is closely linked with social prescribing and personalised care. Further explanation is added here to make this clear.

We added:

“The capacity of arts initiatives to offer personalised rather than prescriptive care and do this early on and prior to mental health problems arising appears to be an advantage that can be used to promote scaling up activities.”

P23, near the bottom: “Expanding the potential use” – this sentence is a statement without substantiation.

Response: This is now substantiated. See edits and additions:

“Exploring the potential use of the model with mixed groups of people, e.g., with different forms of diagnoses or no diagnosis at all (as suggested by one of the NHS managers above: “It doesn’t have to be just a group of people who attract a particular diagnosis; it could just be a mixed group"), appeared to indicate that the work was relevant to a wide range of people needing support with their mental health and wellbeing. Using creative interventions with different groups such as those identifying as neurodivergent was also proposed:

“An increase in LD / ASC + mental health diagnosis would make interventions such as A4B invaluable as an alternative approach.” (Frontline staff, in-person training day 1).”

P24: I think the quote related to Diverse communities is ambiguous and possibly subject to misinterpretation.

Also, what is a ‘south western community’? 

Response: We edited the end of this section in the text.

See:

“... thus, extend therapy provision beyond white populations current provision.”

This was what the participants said. We have edited this as follows: “a south [Asian] community”.

We also added the following text and associated quote in order to strengthen this point:

"while a frontline practitioner argued that the Arts for the Blues intervention is “urgently needed in the black community especially young adult” (Frontline staff, in-person training, day 1).

---

## [Decision Letter · Decision Letter 1]

8 Dec 2023

Developing a Strategy to Scale up Place-Based Arts Initiatives that Support Mental Health and Wellbeing: A Realist Evaluation of ‘Arts for the Blues’

PONE-D-23-11476R1

Dear Dr. Karkou

We’re pleased to inform you that your manuscript has been judged scientifically suitable for publication and will be formally accepted for publication once it meets all outstanding technical requirements.

Kind regards,

Soham Bandyopadhyay

Academic Editor

PLOS ONE

Reviewers' comments:

Reviewer's Responses to Questions

**Comments to the Author**

1. If the authors have adequately addressed your comments raised in a previous round of review and you feel that this manuscript is now acceptable for publication, you may indicate that here to bypass the “Comments to the Author” section, enter your conflict of interest statement in the “Confidential to Editor” section, and submit your "Accept" recommendation.

Reviewer #1: All comments have been addressed

2. Is the manuscript technically sound, and do the data support the conclusions?

Reviewer #1: (No Response)

3. Has the statistical analysis been performed appropriately and rigorously? 

Reviewer #1: (No Response)

4. Have the authors made all data underlying the findings in their manuscript fully available?

Reviewer #1: (No Response)

5. Is the manuscript presented in an intelligible fashion and written in standard English?

Reviewer #1: (No Response)

6. Review Comments to the Author

Reviewer #1: The authors have taken time and care in responding to the reviewer comments. This article will contribute much to the ongoing discussion on this topic.

7. PLOS authors have the option to publish the peer review history of their article (what does this mean?). If published, this will include your full peer review and any attached files.

Reviewer #1: No

---

## [Editor Report · Acceptance letter]

18 Dec 2023

PONE-D-23-11476R1 

PLOS ONE

Dear Dr. Karkou, 

I'm pleased to inform you that your manuscript has been deemed suitable for publication in PLOS ONE. Congratulations! Your manuscript is now being handed over to our production team.

Kind regards, 

on behalf of

Dr. Soham Bandyopadhyay 

Academic Editor

PLOS ONE